# Learning a metacognition for object perception

**Marlene Berke**
Yale Psychology
marlene.berke@yale.edu

**Mario Belledonne**
Yale Psychology
mario.belledonne@yale.edu

**Julian Jara-Ettinger**
Yale Psychology and Computer Science
julian.jara-ettinger@yale.edu

## Abstract

Beyond representing the external world, humans also represent their own cognitive processes. In the context of perception, this metacognition helps us identify unreliable percepts, such as when we recognize that we are experiencing an illusion. In this paper we propose MetaGen, a model for the unsupervised learning of metacognition. In MetaGen, metacognition is expressed as a generative model of how a perceptual system transforms raw sensory data into noisy percepts. Using basic principles of how the world works (such as object permanence, part of infants' core knowledge), MetaGen jointly infers the objects in the world causing the percepts and a representation of its own perceptual system. MetaGen can then use this metacognition to infer which objects are actually present in the world, thereby flagging missed or hallucinated objects. On a synthetic dataset of world states and black-box visual systems, we find that MetaGen can quickly learn a metacognition and improve the system's overall accuracy, outperforming baseline models that lack a metacognition.

## 1 Introduction

Learning accurate representations of the world is critical for prediction, inference, and planning in complex environments [1, 2]. In humans, these representations are generated by perceptual systems that transform raw sensory data, such as light entering the retina, into conceptual representations of the physical space and the objects and agents in it [3, 4]. While human perception is robust and reliable, it nonetheless suffers from rare but compelling errors, such as the Blivet, the lilac chaser, and the peripheral drift illusions. Critically, in all of these cases, people recognize that the faulty representation does not reflect reality and stems instead from an error in their visual system.

Pre-trained systems for object recognition and image classification face a similar challenge: identifying false percepts. After training, these systems can be inflexible and have no general way of identifying when a percept is unreliable or false [1, 5, 6]. As an illustrative example, suppose that an object recognition system observes a 10-frame video of a person riding a motorcycle, and that the system outputs noisy labels for each frame. Suppose than, in some frames, the system incorrectly detects a car that's not actually present. In other frames, it misses the motorcycle or the person. The problem the system faces is to figure out what objects were actually present in the video given these noisy percepts. Put another way, the system has to decide which percepts reflect objects in the external world, and which percepts are merely artifacts of its own processing. We propose that augmenting object recognition and image classification systems with a metacognition—a representation of their own computational processes [7]—can allow these systems to monitor their percepts and flexibly decide when to trust or question proposed representations, much like humans do.

Here we present *MetaGen*, a model for learning a metacognition in an unsupervised context. MetaGen can learn a metacognition for a pre-trained black-box system, requiring only a few percepts from the system. MetaGen does not need access to the internal structure of the system or any performance metrics, meaning that MetaGen can learn a metacognition for a completely black-box system without any external feedback.

Given a set of noisy percepts, we conceptualize learning a metacognition as a joint inference problem over the objects generating the percepts and a representation of the system's performance. Crucially, we make this problem tractable by drawing on two insights from cognitive science: 1) Infants come into the world equipped with a basic form of 'core knowledge' or 'start-up sofware,'thought to be critical for human-like learning [1, 8]. These built-in principles of how the world works constrain the representations of states of the world that infants consider possible (e.g., objects persist in time and move continuously in space) [9]. Similarly, MetaGen constrains the space of representations through a prior distribution over possible world states. 2) Human mental models are often simplified approximations of the content they model, designed for efficient inference and prediction [10, 11]. In the same spirit, metacognition in MetaGen is expressed as a generative model that captures the marginal distributions that approximate a system's performance without modeling the system's exact internal structure and computations.

We apply MetaGen to the problem of identifying what objects are in a scene given a set of percepts. These percepts are drawn from multiple viewpoints or from multiple frames in a video, each processed by the same noisy black-box object recognition system with unknown performance (Figure 1). We formalize a metacognition as a representation of the system's propensity to miss objects that are present and to hallucinate (false alarm) objects that are not present as a function of the object's category. By assuming object permanence (i.e., within each set of percepts, all images include the same objects), our model learns about its own propensity to miss or hallucinate objects in an unsupervised manner. MetaGen can then use this metacognition to identify which objects are likely present in the scene. We test our model using a synthetic dataset where we sample different world states (i.e., scenes with collections of objects) and different visual systems (i.e., visual systems with variable fidelity). We then process these sampled world states through the black-box visual systems to obtain noisy percepts. These percepts are then used as input to MetaGen. We evaluate MetaGen in two ways: 1) by its estimation of the true underlying visual system generating the percepts and 2) by its capacity to flag missed or hallucinated objects, measured in terms of overall accuracy and compared to a set of baseline models.

Overall, our work makes three contributions. First, we present MetaGen, a model for learning a metacognition in an unsupervised manner. Second, we show proof of concept that MetaGen can learn a system's miss and false alarm rates by observing percepts from a few dozen world states. Finally, we show how this learned metacognition enables the model to infer the true world states producing them. For a discussion of related work, please see A.1.

## 2 MetaGen

### 2.1 Problem and solution overview

We first explain the logic of our model in the context of our experimental setup. Consider a black-box classification system that generates percepts for the objects present in a scene (represented as labels from a set of categories). Given a set of object classes $C$ that the system can recognize, the set of possible percepts $X$ is given by the powerset of $C$ (i.e., the set of all subsets of $C$). Similarly, we assume that the space of world states $w \in W$ is given by the powerset of $C$, such that objects are either present or absent in a scene (and thereby assuming object permanence). For instance, in Figure 1, the class of objects $C$ consists of five geometrical shapes. The possible states of the world and the possible percepts consist of all subsets of these five shapes. We define an observation $o = \{x_i\}_{i=1}^{F}$ as a collection of $F$ percepts generated by processing different views of the same world state. In Figure 1, the observation of the first world state contains three percepts ([cone, cylinder], [cone, cylinder, sphere], and [cone]), and the observation of the second world state contains two percepts ([cone, cylinder, cube] and [cone, sphere, cube]).

We define a metacognition as a generative model $v : X \times W \to [0, 1]$ such that $v(x, w)$ is the probability that the visual system would produce percept $x \in X$ when processing world state $w \in W$.

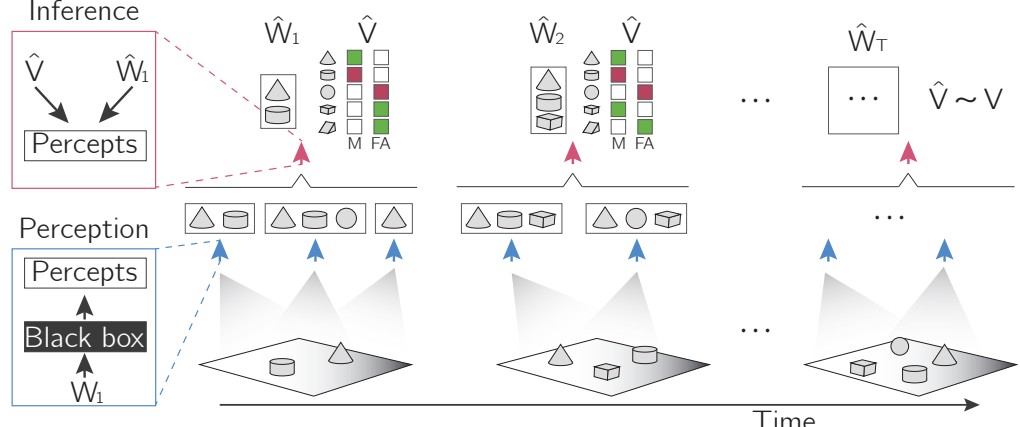

Figure 1: A set of scenes with different objects (bottom row) are processed by a black-box noisy object detection system (blue arrows). MetaGen performs joint inference over the objects in the world states (represented here as a vector of objects $\hat{w}_t$), and a metacognition of the object detection system (represented here as a matrix $\hat{v}$ that captures the visual system's miss rate M and false alarm rate FA for each object category). Colored-in elements of the visual metarepresentation $\hat{v}$ indicate increases (red) and decreases (green) in the estimate ($\hat{v}_i$) of the FA or M for an object category.

Given a collection of observations $\vec{o} = \{o_t\}_{t=1}^{T}$ (i.e., multiple sets of percepts from multiple world states), our goal is to infer $Pr(v, \vec{w}|\vec{o})$, given by

$$Pr(v, \vec{w}|\vec{o}) \propto Pr(v) \prod_{t=1}^{T} Pr(o_t|w_t, v) Pr(w_t) \tag{1}$$

Here we focus on a case where the goal of the metacognitive representation $v$ is to capture the false alarm (false positive) and miss (false negative) rates for each category of objects that could appear in a world state. In this formulation, $v$ is a $C \times 2$ matrix of false alarm and miss rates for the $C$ object categories (see Figure 1 for an example). The framework we propose for learning metacognition is not specific to the choice of simplified representation $v$; it could also apply to a $C \times C$ confusion matrix (where each element at $i, j$ is the probability of mistaking an object of category $i$ for object of category $j$) or to some other representation of $v$.

Figure 1 shows an intuitive explanation of this inference procedure. After the visual system processes the first world state and outputs percepts, MetaGen takes those percepts as input and tries to infer the world state and visual system that caused them. The pattern in the three percepts from the first world state can be explained by inferring that a cone and cylinder were present and that the sphere in the second percept was a false alarm. Because the cone was detected in all three views but the cylinder was only detected in two of them, an appropriate metacognition should decrease its belief that the visual system misses cones and increase its belief that it misses cylinders. Furthermore, the presence of a sphere in the second percept suggests that the false alarm rate for spheres is high. Conversely, the lack of detection of prisms or cubes suggests that the false alarm rate for these shapes is low. Figure 1 visualizes these changes in beliefs.

## 2.2 Generative Model

In MetaGen, the production of percepts is captured through a generative model. If an object $r$ of category $c$ is present in world state $w$, then that object $r_c$ is detected with probability $1 - M_c$ and missed with probability $M_c$. If an object of category $c$ is not present, then that object is hallucinated with probability $FA_c$ and correctly rejected with probability $1 - FA_c$.

$$Pr(r_c \in x) = \begin{cases} 1 - M_c, & \text{if } r_c \in w \\ FA_c, & \text{if } r_c \notin w \end{cases}$$

Thus, which objects are perceived in percept $x$ is a function of the visual system $v$ and the world state $w$.

## 2.3 Inference Procedure

In MetaGen, the posterior, eq. 1, is approximated via Sequential Monte-Carlo using a particle filter [12]. Given $||\vec{o}|| = T$, an estimate of the joint posterior can be sequentially approximated via:

$$Pr(v, \vec{w}|\vec{o}) \approx Pr(\hat{v}^0) \prod_{t=1}^{T} Pr(\vec{o}_t|\hat{v}^t, \hat{w}_t^{\ t}) Pr(\hat{w}_t^{\ t}) Pr(\hat{v}^t|\hat{v}^{t-1}) \qquad (2)$$

where $\hat{v}^T$, is the estimate of $v$, and $\hat{w}_1^{\ T}, \ldots, \hat{w}_T^{\ T}$ is the estimate $w_1, \ldots, w_T$ after $T$ observations. Here the transition kernel, $Pr(\hat{v}^t|\hat{v}^{t-1})$, defines the identity function. The details of the implementation of this inference procedure are left to A.3.

MetaGen can be evaluated in two ways. First, we can test performance change over time as MetaGen learns a metacognition (Online MetaGen). After sufficient observations, the posterior distribution over $v$ stabilizes and MetaGen has a learned metacognition. At this point, we can evaluate the benefit of having a metacognition by fixing the final estimate of $v$ and re-evaluating the world states that caused the observed percepts (Retrospective MetaGen).

# 3 Experiments

To evaluate MetaGen, we created a synthetic dataset of different visual systems processing multiple observations of multiple world states. We then tested MetaGen's ability to infer the underlying visual system and the world states causing the percepts.

**Dataset.** We aim to test whether MetaGen can learn an accurate and useful representation of an artificial visual system. To fully explore the space of possible artificial visual systems, we synthesized a dataset of artificial visual systems with a wide range of probabilities of hallucinating or missing objects. We also synthesized world states (hypothetical collections of objects, summarized as a vector of 1s and 0s indicating the presence or absence of objects). We then generated sparse observations (5-15 per world states; far fewer than what is available on real datasets like videoclips) from these visual system and world states.

Our dataset consists of 35000 randomly sampled visual systems processing multiple views of multiple world states. The details of synthesizing this dataset are left to A.4.

## 3.1 Comparison Models

To better interpret the results of how MetaGen learns a metacognition (Online MetaGen) and how it performs after having learned its metacognition (Retrospective MetaGen), we contrasted our results with two baseline models: Thresholding and Lesioned MetaGen. Thresholding simply concludes that an object was present if it was perceived in more than half of the frames, else, not. For a discussion of other threshold values, please see A.5. Lesioned MetaGen fixes the metacognitive representation of $v$ to the expectation of the prior over $v$. Although it has a metacognitive representation of $v$, that metacognition is neither learned nor updated in light of new observations. Our main models and the comparison models are described more formally in A.5.

## 3.2 Results

In this section, we demonstrate that MetaGen can learn a system's false alarm and hit rates without feedback, and that its improvements in accuracy track with its learning of a metacognition. See A.6 for definitions of the metrics used, and A.2 for results from an example simulation.

Figure 2 shows MetaGen's performance over the 35000 sampled visual systems. Figure 2A shows that Online MetaGen's estimates $\hat{v}_t$ of the false alarm and miss rates per category rapidly approach the true values. In as few as 40 observations, the MSE of $\hat{v}_t$ (FA: 0.0018, M: 0.0033) is less than a third of its initial MSE of $v_{0,\mu}$ (0.011; horizontal dotted line).

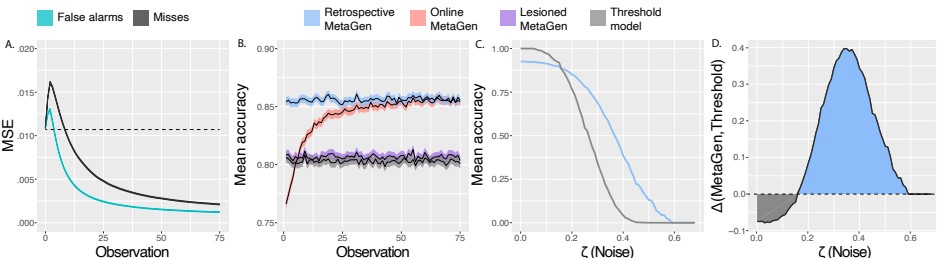

Figure 2: Model performance over 35000 sampled visual systems, each processing 75 world states. **A.** MSE between true and inferred false alarm and miss rates as a function of number of observations. Horizontal dotted line represents MSE from the prior. **B.** Mean accuracy as a function of number of observations. **C.** Accuracy for Retrospective MetaGen and Thresholding as a function of noise in the observation. **D.** Difference in average accuracy between Retrospective MetaGen and Thresholding as a function of noise in the observation.

Figure 2B shows the average accuracy of the models' inferences about world states. The red line shows Online MetaGen's rapid increase in accuracy over the first 40 observations. During these observations, Online MetaGen's accuracy increases from 76.6% to 84.9% (with 85.4% final accuracy after the 75th observation), a percentage substantially higher than chance ($8\%$, see A.7). This increase in accuracy occurs simultaneously with the decline in MSE for the inferred visual system parameters.

After Online MetaGen has completed its final estimate of $v$, Retrospective MetaGen (blue line; Figure 2B), revises its beliefs about the world states by conditioning on that final estimate. Although its estimate of $v$ was based on percepts of those world states, the model never received feedback (access to the ground-truth of those world states). Retrospective MetaGen (red; 85.6% average accuracy), consistently outperforms Thresholding (light gray; 80.3% accuracy) and the Lesioned MetaGen model (purple; 80.7%) by about 5%. Together, these results show that MetaGen's ability to infer the true world states was not due to the high fidelity of the percepts (as it outperformed the Threshold model) or due to merely having a metacognitive representation (as it outperformed Lesioned MetaGen). Instead, our results show that MetaGen's improved accuracy was due to its ability to learn the content of the metarepresentation in an unsupervised manner.

Retrospective MetaGen especially outperforms Thresholding on noisy percepts. Figure 2C shows the average accuracy of Retrospective MetaGen and Thresholding as a function of noise $\zeta$ (using a rolling window such that each point shows average accuracy on the $[\zeta - .05, \zeta + .05]$ range. See A.6 for a formal definition of perceptual noise, $\zeta$). At low noise levels ($\zeta \in [0, 0.15]$), Thresholding reaches near ceiling performance because, by definition, the majority of the percepts are accurate. As percepts become noisy ($\zeta \in [0.15, 0.59]$), both models decline in accuracy, but MetaGen declines more gradually, outperforming the Threshold model. This implies that MetaGen gains accuracy over Thresholding by sometimes rejecting objects that appear on the majority of percepts and choosing to include objects that were only present in a minority of percepts. Figure 2D shows the difference in average accuracy between MetaGen and Thresholding as a function of perceived noise. Metagen reaches peak advantage over Thresholding at a noise level of $\zeta = 0.34$, with 58.9% accuracy, while Thresholding is at 19.2% accuracy.

Given that MetaGen drastically outperforms Thresholding for most noise levels, it may seem surprising that MetaGen's overall accuracy is only 5% above that of Thresholding. Note, however, that percepts are biased toward having low levels of noise. At such low noise levels, Thresholding can reach ceiling performance and obscure MetaGen's drastic success over Thresholding on noisy percepts.

## 4  Discussion and Conclusion

Here we proposed MetaGen, a model for learning a metacognition in an unsupervised context. Given a set of observations generated by a black-box classification system with unknown performance, MetaGen performs joint inference over a meta-representation of the system and over the objects causing the observations. Using a large synthetic dataset of black-box visual systems, we showed

that with as few as 40 observations, MetaGen can infer a system's propensity to false alarm and miss objects. MetaGen can use this metacognition to flag and correct errors from the classification system, improving the system's overall accuracy.

Although our analyses were based on a synthetic dataset, we have demos showing that MetaGen can infer stable object representations from the outputs of real-world artificial visual systems, like Detectron2 [13]. These demos can be found on the project's GitHub page (link).

More broadly, our work points to a new direction for how we think about improving AI systems. When an AI system's representations of the world are noisy, a typical approach is to try to directly improve those representations of the world (i.e. change the visual system's architecture or training set so as to directly reduce the false alarm and miss rates). Our work suggests a complimentary approach: learning representations of the system itself. Representing how an AI system builds its representations of the world can facilitate compensation for the noisiness of these representations. Our work shows how this can be achieved without feedback, by grounding this learning process in basic assumptions about the world (object permanence, in our case). This model offers a proof-of-concept of such an approach, and highlights the importance of metacognition in humans and machines.

### 4.1 Acknowledgments

We thank Laurie Paul, Ilker Yildirim, and Flora Zhang for helpful discussions. This work was supported by a Google Faculty Research Award and by the Center for Brains, Minds, and Machines NSF-STC award CCF-1231216.

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

# A    Appendices

## A.1    Related Work

**Metacognition in AI.** Previous work has argued for the importance of metacognition for machine learning and AI [14]. Models that use metacognition during learning have shown promise for improving classification accuracy [15, 16]. This work has focused on engineering an inflexible metacognition to guide a system's learning. In this paper, we focus on a complimentary problem: developing a model for learning a metacognition of any black-box system.

**Computational cognitive science.** Our core idea—learning a metacognitive model of a perceptual system—is inspired by research in cognitive science showing that human reasoning is structured around mental models of the physical world [11, 17], of the social world [18, 19], and of ourselves [20, 21]. A critical idea in this line of research is that mental models do not need to capture the true data-generating process. Instead, these models are often approximations that are broadly accurate, but simplified to make inference and reasoning more tractable than would be otherwise possible [17, 18]. This type of work aims to model human intuitive theories. Unlike this work, our work does not aim to model people's metacognition. Our goal is instead to test, in a machine-learning and AI context, whether the type of mental models that humans use to reason about themselves and others can serve as a fruitful approach for learning how to represent complex black-box systems.

Our work is also related to computational models of human core knowledge [22, 23]. Although we use infant-like core knowledge to support learning models about oneself, our focus is not on core knowledge per se.

**Uncertainty-aware AI.** The spirit of our work relates more closely to uncertainty-aware AI. This work focuses on building end-to-end systems that express uncertainty in their inferences [24–26]. Our work focuses instead on a related problem: how can you learn a model of uncertainty over a pre-trained, black-box system? These two approaches complement each other. In humans, meta-cognitive uncertainty supplements the intrinsic uncertainty in visual perception. For example, when estimating the number of dots in an array, people experience a basic, perceptual kind of uncertainty integrated with a higher-level, conceptual kind of uncertainty. [27, 28]).

## A.2    Example Simulation

Here, we show an example simulation from our experiment. Figure 3A shows a sampled set of 75 world states. Each row represents one of the five objects, each column a world state, and color indicates the presence or absence of an object (light blue indicating presence). Figure 3B shows the observations generated by these 75 world states when passed through a noisy black-box object detection system (color indicating proportion of percepts in which the object was detected). Figures 3C and 3D show the inferred realities obtained by the Thresholding and MetaGen models, respectively. Figures 3E and 3F show the inferred world states color-coded by whether object were correctly detected, false alarmed, or missed. In this simulation, the visual system has high false alarm rates for categories C (sphere) and D (cuboid), leading to a high proportion of false alarms for these categories. Thresholding takes these noisy percepts at face value, resulting in the incorrect conclusion that nearly every world state contains objects of category C and D. MetaGen, by contrast, learns the visual system's bias toward false alarming these categories and is able to drastically reduce the number of false alarms, at the expense of introducing a few misses.

## A.3    Inference Procedure

We sequentially approximated the joint posterior given in eq. 2 using a particles filter with 100 particles.

We implemented rejuvenation using a series of Metropolis-Hastings MCMC perturbation moves over $\hat{v}$. The proposal function is defined as a truncated normal distribution with bounds $(0, 1)$:

$$\hat{v_i}^{t'} \sim \mathcal{N}(\mu = \hat{v_i}^{t}, \sigma^2 = 0.01) \tag{3}$$

where $v_i$ an element in the matrix $v$. A proposal is accepted or rejected according to the Metropolis-Hastings algorithm [29]. Each element in $v$ is rejuvenated separately and in randomized order.

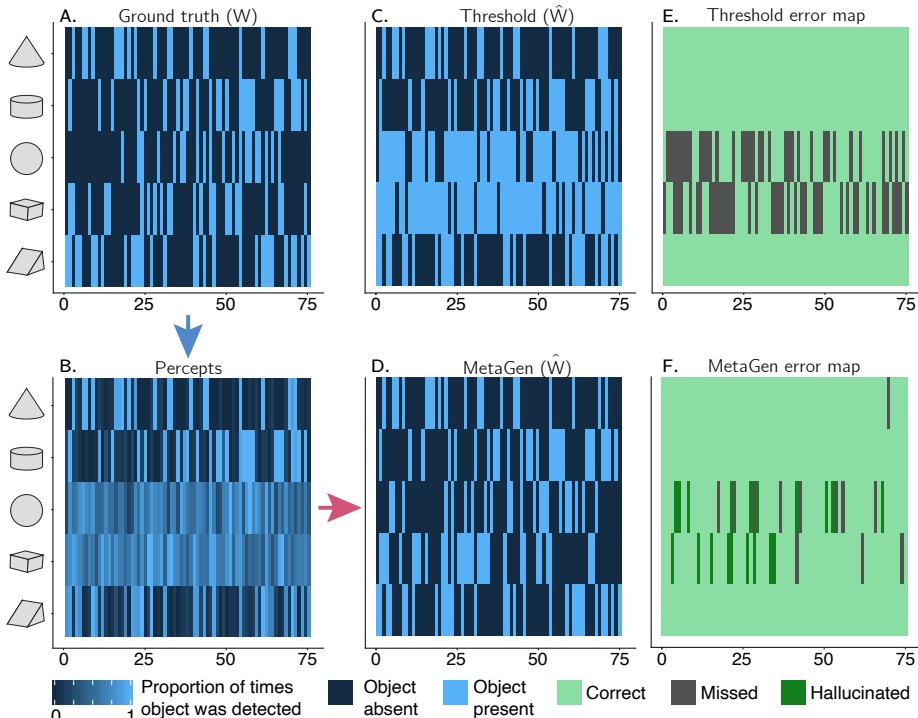

Figure 3: Example simulation from our experiment. **A.** Ground-truth world states. Each column represents a sampled world state (75 world states total). The presence or absence of an object is coded as light and dark blue, respectively. **B.** Percepts generated by a visual system processing 5-15 viewpoints per world state. Color indicates proportion of viewpoints from which an object was detected (lighter blue indicating higher proportion). **C** Inferred world states obtained by thresholding the percepts from panel B. **D.** Inferred world states inferred (retrospectively) by MetaGen from the percepts from panel B. **E-F** Thresholding and MetaGen's error maps. Light green indicates correct inferences, grey indicates objects that the model missed, and dark green indicates objects that the model false alarmed.

We then take the expectation of the marginal distribution by averaging across unweighted particles: $\hat{v}^T_\mu = E[\hat{v}^T | \vec{o}] = \frac{1}{M} \sum_{m=1}^{M} \hat{v}^T_m$, where $m$ indexes the particles. Given $\hat{v}^T_\mu$, the posterior predictive distribution is defined as:

$$Pr(\hat{\vec{w}} | \vec{o}, v = \hat{v}^T_\mu) \propto Pr(v = \hat{v}^T_\mu) \prod_{t=1}^{T} Pr(o_t | \hat{w}_t, v = \hat{v}^T_\mu) Pr(\hat{w}_t) \qquad (4)$$

This posterior predictive distribution can then be used to make better inferences about world states. These world states could be new ones $w_{T+1}, ...,$ or they could be the world states $w_1, ..., w_T$ already used to estimate $v$. Retrospective MetaGen does that latter: conditioning on $\hat{v}^T_\mu$, Retrospective MetaGen infers the world states $w_1, ..., w_T$ from the posterior predictive distribution. When inferring these world states, we take the MAP of the unweighted particles.

We implemented our generative model and inference procedure in the Julia-based probabilistic programming language Gen [30].

## A.4 Synthesizing the Dataset

Here we discuss how we synthesized the dataset for evaluating MetaGen.

In this context, the visual system, $v$ can be represented as a $5 \times 2$ matrix of false alarm and miss rates (see Figure 1). Each visual system was generated by drawing ten independent samples from a beta distribution, $\sim B(\alpha = 2, \beta = 10)$. This distribution allows us to sample visual systems with variable

error rates (mean value = .17) while maintaining a low probability of sampling visual systems that produced false alarms or misses more often than chance (0.005 chance of sampling values above 0.5; 0.06 chance that complete sampled visual system has at least one false alarm or miss rate above 0.5). The wide range of these false alarm and miss rates encompasses those of state-of-the-art artificial visual systems.

For each visual system, we sampled 75 world states. A Poisson distribution $N \sim Poisson(\lambda = 1)$ truncated with bounds $[1, 5]$ determined the number of objects in a world state. The object categories were samples from a uniform distribution. Each world state was a hypothetical collections of objects, summarized as a vector of 1s and 0s indicating the presence or absence of each category of objects. For each world state we used the visual system to synthesize the $5 - 15$ percepts (number sampled from a uniform distribution), producing a total of $375 - 1125$ percepts per visual system. Inferences about the false alarm and miss rate of each object are independent, and we thus considered situations with only five types of objects.

## A.5 Comparison Models

A model needs to map a collection of percepts to a likely world state. One simple way to recover the world state $w$ causing observation $o = x_1, \ldots, x_F$ is to threshold the percepts—an object $c$ was present if and only if it appeared in at least half (0.50) of the observations. We call this model Thresholding.

It is possible that the Thesholding baseline model would perform better with a different threshold value. Fitting the threshold value involves comparing the model's outputs to ground-truth world states. Although this Fitted Thresholding model, unlike MetaGen, has access to ground-truth, we nevertheless compared the two. We found that, even using the best-fitting threshold value (0.54), Retrospective MetaGen still outperformed this Thresholding model with an overall average accuracy of 85.6% compared to Fitted Thresholding's 83.1%. Even granting this baseline model access to ground-truth, it could not outperform our unsupervised MetaGen model.

To test whether learning a metacognition improves inferences about the world state, we compare MetaGen with and without the metacognitive learning. We call MetaGen without learned metacognition Lesioned MetaGen. Like the other MetaGen models, Lesioned MetaGen has a metacognitive representation of $v$ and uses an assumption of object permanence to infer the world states causing the percepts. Lesioned MetaGen, however, does not learn or adjust the content in its meta-represetation $v$ based on the observed percepts. Formally, Lesioned MetaGen assumes that the false alarm and miss rates for every category are the MAP of the beta prior over false alarm and miss rates, call it $\hat{v}_{0,\mu}$. Lesioned MetaGen then uses the same particle filtering process described in A.3, except that it conditions on $\hat{v}_{0,\mu}$ instead of $\hat{v}_{T,\mu}$.

We also compare two variations of MetaGen with learning. Online MetaGen performs a joint inference over $v$ and $\vec{o}$ online, as observations are presented sequentially. This model allows us to evaluate how MetaGen's inferences improve as a function of the observations it has received. We name this model of online, observation-by-observation inference Online MetaGen.

After having received all $T$ observations, MetaGen could retrospectively re-infer the world states causing the $T$ observations. This model, Retrospective MetaGen, re-infers the world states that caused its observations conditioned on its estimate $\hat{v}_{T,\mu}$, as described in A.3.

Online MetaGen lets us interpret how MetaGen learns a metacognition, and Retrospective MetaGen lets us test how MetaGen performs after having learned that metacognition. Thresholding and Lesioned MetaGen serve as baseline models for comparison.

## A.6 Metrics Used

To measure how well MetaGen learned a metacognition, we calculated the mean squared error (MSE) between the inferred visual system $\hat{v}$ and the true $v$ generating the percepts, given by

$$\text{MSE} = \frac{1}{2|C|} \sum_{c \in C} \left( (FA_c - \hat{FA_c})^2 + (M_c - \hat{M}_c)^2) \right)$$ (5)

where $C$ is the the set of object classes.

To measure MetaGen's capacity to infer the objects causing the percepts, we computed world-state accuracy, defined as 1 if $\hat{w} = w$ (i.e., the model inferred the exact set of object that are present) and 0 otherwise. This produces one accuracy score per world state, for a total of 75 accuracy scores per visual system. We average these accuracy scores to get the average accuracy. For this baseline measure, expected accuracy from the prior (without learning a metacognition) is 8% (see A.7).

To analyze MetaGen's accuracy as a function of noise in the percepts, we computed the average noise of an observation $o_t$ as

$$\zeta_t = \frac{1}{|o_t||C|} \sum_{c \in C} \sum_{x \in o_t} |\mathbb{1}_{w_t}(c) - \mathbb{1}_x(c)| \tag{6}$$

where $C$ is the set of object classes, $o_t$ is the collection of percepts generated from world state $w_t$, $\mathbb{1}_{w_t}(c)$ is an indicator for whether an object of class $c$ is in $w_t$, and $\mathbb{1}_x(c)$ is an indicator for whether an object of class c is in percept $x$.

### A.7 Expected Accuracy of Guessing World States

Here, we calculate the expected accuracy of a model that guesses world states by sampling from the prior over possible world states. Call $d(n)$ the density of a Poisson distribution truncated with bounds $[1, 5]$ and with $\lambda = 1$. If $n$ is the number of objects in a world state then the probability of guessing a world state correctly by chance is $\sum_{n=1}^{n=5} d(n) * d(n) / \binom{5}{n} = 0.0774$. Any model that performs significantly above 8% accuracy is above chance.

We want to calculate the expected accuracy of a model that guesses world states by sampling from the prior. Call $d(n)$ the density of a Poisson distribution truncated with bounds $[1, 5]$ and with $\lambda = 1$.

If a ground-truth world state has n objects, then the probability of guessing the correct number of objects by sampling from the prior is $d(n)$. The probability of guessing the correct combination of n objects out of the 5 possible objects is $\frac{1}{\binom{5}{n}}$. So, given that a ground-truth world state has n objects, the probability of guessing that world state correctly by sampling from the prior is $d(n)/\binom{5}{n}$.

We want the expectation of guessing correctly when the ground-truth n is unknown. We must sum, over all n, the probability of guessing correctly, weighted by the probability that the ground-truth world state has n objects. The probability that the ground truth has n objects is the prior over n, $d(n)$. So we obtain $\sum_{n=1}^{n=5} d(n) * d(n)/\binom{5}{n}$.

