# OpenReview forum: "Learning a metacognition for object perception"
_NeurIPS.cc/2020/Workshop/SVRHM — SVRHM@NeurIPS Poster_

### Official Review · AnonReviewer3 · 2020-10-27
**Review: Learning a metacognition for object perception**

**Rating:** 8
**Confidence:** 4

**Review:**


This paper presents MetaGen, an unsupervised model framework that learns a metacognition of object perception for a given visual system. MetaGen learns the propensity of a visual system to hallucinate or miss a given object in a given scene by comparing multiple views of the scene, and infer from this which object are truly present.

Pros:
- This paper asks a very interesting question - how can integrating over multiple views of the world improve the quality of an object detection system? -  and ties this question to important cognitive processes by making the link to metacognition. The question is well-motivated with cognitive and computational literature, and the paper is extremely well written. I found the explanations largely to be clear and measured, and thought the authors made lots of good efforts to make sure the reader is on board at each step of the way.

- MetaGen is compared against two alternative (Threshold model and Lesioned MetaGen). I thought these alternative models were very informative and well-chosen. On a side note, this led me to wonder - how well does a metacognition learned over one visual system (or world state) perform on another visual system (or world state)? This could be another interesting benchmark, although I don’t believe the authors need to include that here, I’m just curious.


Cons:
- I was confused by the description of the dataset. Specifically, I struggled to understand what “synthesizing a dataset of artificial visual systems with a wide range of probabilities of hallucinating or missing objects” entailed. Does this mean that you generated many “percepts” for a given black-box model, such that they added up to a given false alarm and miss probabilities, and that over many black boxes, you covered a large range of miss and false alarm probabilities? What does a “world state” consists of - is this a purely theoretical concept at this stage (i.e. the world is summarized as a vector of 0s and 1s), or are these visual stimuli, either of simple geometric shapes on a field, similar to figure 1, or rich-real world images? This section was quite dense (no doubt because of constraints on the length of the paper), but some more concrete details here would have been helpful.

- The results of the Retrospective MetaGen are useful, as they serve to show that once the metacognition is learned, it leads to better understanding of world states that the model had previously encountered. However, there is circularity in that analysis, since this is operating over views that were used in the learning. This should either be addressed when discussing what the reader should conclude from the Retrospective results, or the authors should have held-out views to test on after learning is complete.


some typos:
- extra “the” in line 106
- typo in figure 2 caption: “ratse” instead of rates

---

### Official Review · AnonReviewer2 · 2020-10-29
**A way to incorporate object permanence into a model visual system using unsupervised learning**

**Rating:** 7
**Confidence:** 3

**Review:**

The fundamental premise of the work is that the performance of artificial visual systems can be improved by making them sensitive to object permanence, i.e. the sense that the identities of objects in a given world are constant even though the incoming visual information can be different based on viewpoint. A neat method is presented to exploit this knowledge through an algorithm (MetaGen) that can piggyback on a black-box visual system. At its core, the algorithm is a Bayesian-based approach that dynamically manipulates the prior belief that an object may be present in a world given the incoming visual information. The prior is learnt in an unsupervised manner, based on the outputs of the black box visual system.

The problem setup and framework of analysis are well-described. The methods used in the paper are described clearly. A minor edit is recommended in final section of this review.

Pros:
1. The algorithm can be used universally to improve any visual system that can output the likelihood of a percept sampled from a powerset of possible object combinations.
2. Learns the FA and miss rates rapidly for the toy example with 5 object classes in ~40 trials.
3. The algorithm improves the performance significantly over a naïve thresholding model, especially when the visual system's percepts are noisy (deviates more often from the ground truth).

Comments/ Significance:
Object permanence is one form of prior knowledge that influences our response, but as acknowledged, there are other forms of metacognition that also play a role in visual perception. Humans have many kinds of priors like size, co-occurrence, illumination direction and orientation. This paper suggests that these priors can simply be modeled as biases in responses, either by reducing the threshold to reduce misses or by increasing the threshold to reduce false positives. In that sense, the Threshold model is constrained to move along a specified bias contour on an Iso-Bias contour map in an ROC diagram, whereas MetaGen adds another degree of freedom allowing a change in response bias.  While all of these notions are standard, the novelty of this work is to use a toy dataset that contains multiple viewpoints of a given world, and to nudge a system towards producing the same percept (set of detected objects) for these viewpoints.

Suggested Edit:
Line 73: Present or absent

---

### Official Review · AnonReviewer1 · 2020-11-01
**Review of "Learning a metacognition for object perception"**

**Rating:** 7
**Confidence:** 4

**Review:**

Here the authors present MetaGen, a procedure that learns misses and false alarm rates of classification algorithms in an unsupervised manner, and uses them to decide when a classifier has either missed or hallucinated a particular percept, leading to more reliable and stable estimates of observed object categories. The topic is interesting and novel, and the paper is for the most part clear and well-presented. There are a couple of issues that the authors should consider for a final version of the paper/poster.

- First, the very general introduction and links to cognitive science literature make the paper interesting for a wider audience, but at the same time they distract from the more specific goal of the presented procedure. I could only understand what the authors were attempting in practice after following their link to the GitHub page and seeing that what MetaGen does is to infer stable object representations from the output of an object detection/classification algorithm working over video, which tends to switch labels for the presented objects in different frames. After seeing that, much of what is done in the paper is easier to understand (e.g., the choice of a simple threshold algorithm as a control procedure). I would recommend that the authors clarify that this is the specific problem that MetaGen solves. Admittedly, this is much less attractive than proposing "a framework for learning a metacognition", but it would make the paper/poster easier to read/follow.

- A second important issue is that several choices seem arbitrary and make it difficult to evaluate to what extent MetaGen has a better performance than the comparison models in all conditions, rather than those that are presented in the paper. The clearest example is the choice of threshold value. One can see in Figure 3 that the main difference between the threshold model and MetaGen is that the former favors misses over hallucinations, while the latter strikes a better balance between the two. A different threshold value here would have produced better performance for the threshold model. For example, one could set the threshold individually for each object class, based on the observed base rate of percepts across all observations. This could improve performance and make it similar to that of MetaGen, without having to deal with the computational machinery and burden of approximate inference through particle filters. Beyond that specific possibility, the point is that the threshold is chosen arbitrarily, and it is not clear whether the choice is particularly beneficial for MetaGen in the comparison.

- Another rather arbitrary choice is for the parameters of the beta distribution that generated the synthetic dataset. It is not clear whether the error rates from state-of-the-art computer vision algorithms are captured by this distribution. Although the authors indicate that the wide range of FA and Miss rates encompasses those of artificial visual systems, they don't provide evidence/data/references to back up this assertion, and one would like to not only "encompass" such rates but actually match them.

- Finally, the choice of $\lambda$ for the Poisson distribution also seems arbitrary, and in this case unrealistic as it assumes that most real-world examples would involve presentation of one or a few objects in a scene.

- Some notation choices make things confusing. $p$ and $P$ are used to represent percepts rather than probabilities, but then in line 83 $p$ is actually used to represent a probability, and in Eq. 1 $Pr$ is used in that way. I think that the authors should reserve $p$ and $P$ for probabilities and use something else for percepts.

- It is not clear what $j$ is at the end of Eq 1.

In sum, the final paper/poster would benefit from either less arbitrary choices for parameter values or, even better, an exploration of many parameter values for the beta distribution, the Poisson distribution, and thresholds. In addition, inclusion of a description of the more specific problem solved by MetaGen in the introduction section would make the paper/poster easier to understand.

---

### Public Comment · ~Marlene_Berke1 · 2020-11-25
**Response to reviewers**

Dear Reviewers,

Thank you for your thoughtful comments and valuable suggestions. We think that our paper is better as a result of your feedback! We really appreciate your time and attention.

Reviewer 1 brought up the possibility of using other threshold values for the Thresholding baseline model. We added a paragraph to the A.5 Comparison Models section with results showing that MetaGen outperforms Thresholding even when the threshold value is fit by comparing model outputs to ground-truth world states.

In response to Review 1's other comments, we have removed the language about building a "framework," and have expanded the second paragraph of the introduction to make our goals clearer. We also clarified our contributions and claims about metacognition by adding a final paragraph to the discussion section. We also removed the claim that the beta distributions over false alarm and miss rates capture the error rates of state-of-the-art artificial visual systems.

We'd like to thank R2 for their insightful discussion of response biases. That was a nice framing of the problem that we might use in future work.

Reviewer 3 pointed out some parts that were unclear. To clarify the Dataset section, we added some explanation to the Datasets section and to appendix A.4. We also acknowledged that Retrospective MetaGen's accuracy is assessed on the same percepts that were used to infer its metacognition $v$, and explain that that is permissible because MetaGen never received feedback on those percepts.

Minor edits: We also now use $x$ instead of $p$ for percepts and removed the errant $j$ in equation 1, and corrected the typos in line 73, 106, and the caption of Figure 2.

Thank you!

---

### Decision · Program_Chairs · 2020-11-02

Accept (Poster)